# Electrical and Structural Properties of All-Sputtered Al/SiO$_2$/$p$-GaN MOS Schottky Diode

**Tran Anh Tuan Thi** [1,*] , **Dong-Hau Kuo** [2,*] , **Phuong Thao Cao** [3], **Pham Quoc-Phong** [3], **Vinh Khanh Nghi** [3] **and Nguyen Phuong Lan Tran** [4]

1   School of Basic Sciences, Tra Vinh University, Tra Vinh 87000, Vietnam
2   Department of Materials Science and Engineering, National Taiwan University of Science and Technology, Taipei 10607, Taiwan
3   School of Engineering and Technology, Tra Vinh University, Tra Vinh 87000, Vietnam; cpthao@tvu.edu.vn (P.T.C.); phongpham@tvu.edu.vn (P.Q.-P.); nghivinhkhanh@tvu.edu.vn (V.K.N.)
4   School of Engineering and Technology, Can Tho University, Can Tho 94000, Vietnam; tnplan@ctu.edu.vn
*   Correspondence: thitrananhtuan@tvu.edu.vn (T.A.T.T.); dhkuo@mail.ntust.edu.tw (D.-H.K.); Tel.: +886-2-2730-3291 (D.-H.K.)

**Abstract:** The all-sputtered Al/SiO$_2$/$p$-GaN metal-oxide-semiconductor (MOS) Schottky diode was fabricated by the cost-effective radio-frequency sputtering technique with a cermet target at 400 °C. Using scanning electron microscope (SEM), the thicknesses of the electrodes, insulator SiO$_2$ layer, and $p$-GaN were found to be ~250 nm, 70 nm, and 1 μm, respectively. By Hall measurement of a $p$-Mg-GaN film on an SiO$_2$/Si (100) substrate at room temperature, the hole's concentration ($N_p$) and carrier mobility (μ) were found to be $N_p = 4.32 \times 10^{16}$ cm$^{-3}$ and μ = 7.52 cm$^2 \cdot$V$^{-1} \cdot$s$^{-1}$, respectively. The atomic force microscope (AFM) results showed that the surface topography of the $p$-GaN film had smoother, smaller grains with a root-mean-square (rms) roughness of 3.27 nm. By $I$–$V$ measurements at room temperature (RT), the electrical properties of the diode had a leakage current of ~$4.49 \times 10^{-8}$ A at −1 V, a breakdown voltage of −6 V, a turn-on voltage of ~2.1 V, and a Schottky barrier height (SBH) of 0.67 eV. By $C$–$V$ measurement at RT, with a frequency range of 100–1000 KHz, the concentration of the diode's hole increased from $3.92 \times 10^{16}$ cm$^{-3}$ at 100 kHz to $5.36 \times 10^{16}$ cm$^{-3}$ at 1 MHz, while the Fermi level decreased slightly from 0.109 to 0.099 eV. The SBH of the diode at RT in the $C$–$V$ test was higher than in the $I$–$V$ test because of the induced charges by dielectric layer. In addition, the ideality factor ($n$) and series resistance ($R_s$) determined by Cheung's and Norde's methods, other parameters for MOS diodes were also calculated by $C$–$V$ measurement at different frequencies.

**Keywords:** MOS Schottky diode; SBH; $I$–$V$ measurement; $C$–$V$ measurement; Cheung's and Norde's methods

## 1. Introduction

GaN-based semiconductor materials are currently of interest for the fabrication of electronic devices such as the metal-semiconductor (MS) and MOS Schottky diodes, light-emitting diodes (LEDs), photo-detector, metal-oxide-semiconductor field-effect transistors (MOSFETs), and heterojunction field-effect transistors (HFETs). [1–5]. Previous studies created the thin, high-quality insulator layer between the metal and semiconductor that is used to create a metal-oxide-semiconductor (MOS) structure, which was an important factor for the high-performance of MOS devices [6–10]. Researchers investigated the contact of MOS layers via various approaches, e.g., Al/HfO$_2$/$p$-Si [7], Pt/oxide/$n$-InGaP [10], Pt/SiO$_2$/$n$-InGaN [11], Pd/NiO/GaN [12],

Au/SiO$_2$/*n*-GaN [13], Au/SnO$_x$/*n*-LTPS/glass [14], Pt/SiO$_2$/*n*-GaN [6,15], Pt/Oxide/Al$_{0.3}$Ga$_{0.7}$As [16], Pd/HfO$_2$/GaN [17], and Al/SnO$_2$/*p*-Si (111) [18]. Due to the presence of the oxide layer, several parameters can be applied to improve the characteristics of electronic devices. Bengi et al. reported the parameters of the Al/HfO$_2$/*p*-Si MOS device, which was tested by *C–V* measurement. Their SBHs were shown from 0.17 to 0.98 eV, in the temperature range 300–400 K [7]. Karadeniz et al. investigated the Al/SnO$_2$/*p*-Si (111) diode using spray deposition. The MOS diode showed a Schottky barrier height (SBH) of 0.52 V, an ideality factor of 2.4, and series resistance of 66 Ω [18]. Liu et al. studied the influence of hydrogen adsorption on the Pd/AlGaN-based MOS diode with SiO$_2$ passivation [19]. Their SBHs were reduced from 0.98 to 0.75 eV under exposure to a 1% H$_2$/air gas.

In this study, the radio-frequency (RF) reactive sputtering technique was used to design the Al/SiO$_2$/*p*-GaN MOS Schottky diode because of advantages such as low deposition temperature, low cost, and safety [3,6,11]. With the support of the parameters and using the RF technique, our diode was fabricated below 400 °C. The characteristics of the MOS Schottky diode were tested using *I–V* and *C–V* measurements. The parameters of the diode were calculated by thermionic emission (TE) mode using Cheung's and Norde's methods.

## 2. Materials and Methods

Figure 1 shows the modeling of the Al/SiO$_2$/*p*-GaN MOS Schottky diode based on *p*-GaN film. First, for the Schottky contacts, an Al layer was sputtered on an SiO$_2$/Si (100) substrate at 200 °C for 20 min using a pure Al (99.99%) target, and RF power of 80 W. To construct the MOS Schottky diodes, an interlayer between Al and *p*-GaN was designed by depositing SiO$_2$. The SiO$_2$ film was sputtered on an Al/SiO$_2$/Si (100) substrate at 100 °C for 10 min using a quartz target. The RF power remained at 80 W and the Ar atmosphere at a flow rate of 5 sccm. Second, the Mg-GaN films were deposited onto SiO$_2$/Al/SiO$_2$/Si (100) and SiO$_2$/Si (100) substrates at 400 °C for 40 min. The RF power of was kept at 150 W with a gas mixture of Ar and N$_2$ and a flow rate of 5 sccm for each. The 2-inch Mg-GaN target had an [Mg]/([Ga] + [Mg]) molar ratio of 10% and was made via hot pressing. Finally, a Pt-Omhic contact with a size of 1 mm$^2$ was deposited, at 200 °C for 20 min, with a pure Pt (99.99%) target using a stainless mask.

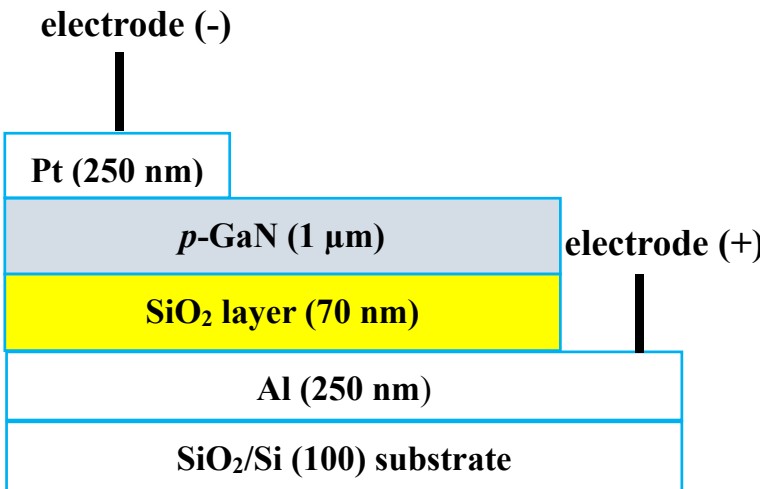

**Figure 1.** The modeling of the as-deposited Al/SiO$_2$/*p*-GaN MOS Schottky diode.

The composition analysis and surface topographies of the *p*-GaN films were determined via SEM and EDS (JSM-6500F, JEOL, Tokyo, Japan), AFM (Dimension Icon, Bruker, Tokyo, Japan). The hole's concentration ($N_p$) and the mobilities (μ) of the *p*-GaN film were calculated by Hall measurement (HMS-2000, Ecopia, Tokyo, Japan). The *I–V* and *C–V* measurement of the MOS Schottky diode were tested using a semiconductor device analyzer (Agilent, B1500A, Santa Clara, CA, USA) at RT. All the

parameters of the MOS Schottky diode were considered by thermionic emission (TE) mode using Cheung's and Norde's methods.

## 3. Results and Discussion

### 3.1. Structural and Surface Morphological Characteristics

By Hall measurement of the *p*-Mg-GaN film on the $SiO_2$/Si (100) substrate at RT, the hole's concentration ($N_p$) and carrier mobility (μ) were found to be $N_p$ = 4.32 × $10^{16}$ $cm^{-3}$ and μ = 7.52 $cm^2 \cdot V^{-1} \cdot s^{-1}$, respectively. Using SEM, the thicknesses of both the electrodes and the $SiO_2$ layer were found to be 250 and 70 nm, respectively.

Figure 2a shows the SEM surface morphologies of the *p*-GaN films sputtered on the $SiO_2$/Si (100) substrate. With EDS analysis results, the ratio of [Mg]/([Ga] + [Mg]) was 10.2% for the *p*-GaN film. This indicated that the *p*-Mg-GaN film deposited at 400 °C with up to 10% Mg displayed continuous smoothness without cracks and pores. The inset shows a cross-sectional image, with a thickness of 1 μm for the *p*-GaN film. Figure 2b shows the surface topography of the as-deposited Mg-GaN films on the $SiO_2$/Si (100) substrate tested by AFM measurement. The surface topography showed smoother and smaller grains and the root-mean-square (rms) roughness of the films was found to be 3.27 nm. The EDS compositions, SEM surface morphologies and XRD patterns of the *p*-Mg-GaN film obtained with cermet targets at different Mg contents can be found in our previous works [20,21]. The positive surface conditions of Mg-GaN layer together with the insulator $SiO_2$ layer were the important factors for determining the electrical properties of the MOS Schottky diodes.

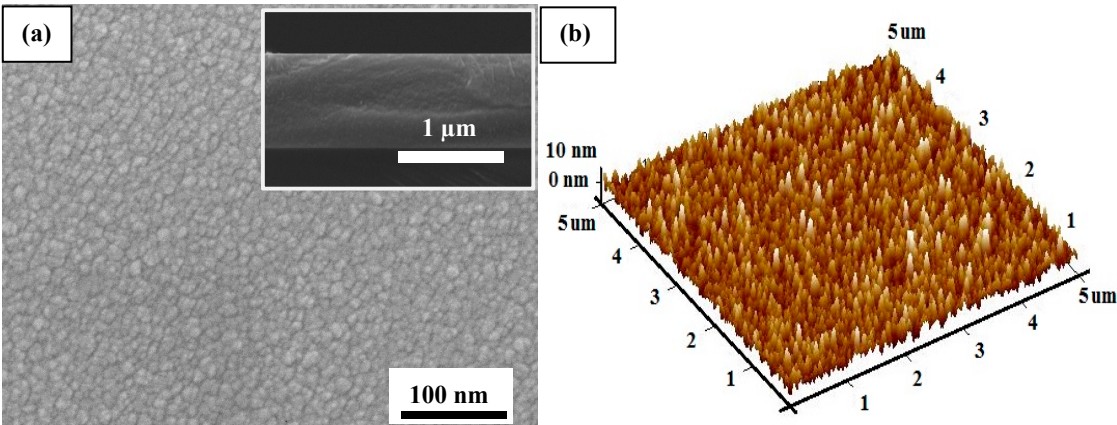

**Figure 2.** (**a**) SEM surface image and (**b**) three-dimensional AFM topographies of the *p*-GaN film deposited on the $SiO_2$/Si (100) substrate. The inset is the cross-sectional image of the *p*-GaN film.

### 3.2. Current–Voltage (I–V) Characteristics

Figure 3a displays the *I–V* plot of the Al/$SiO_2$/p-GaN MOS Schottky diode measured at RT. The Figure 3b shows the ln*I–V* semilogarithmic view of the diode. From the *I–V* data, tested with a voltage range of (−6 V; +6 V) and a leakage current of −1 V, the turn-on voltage of the diode was determined to be ~4.49 × $10^{-6}$ A/$cm^2$ and 2.3 V.

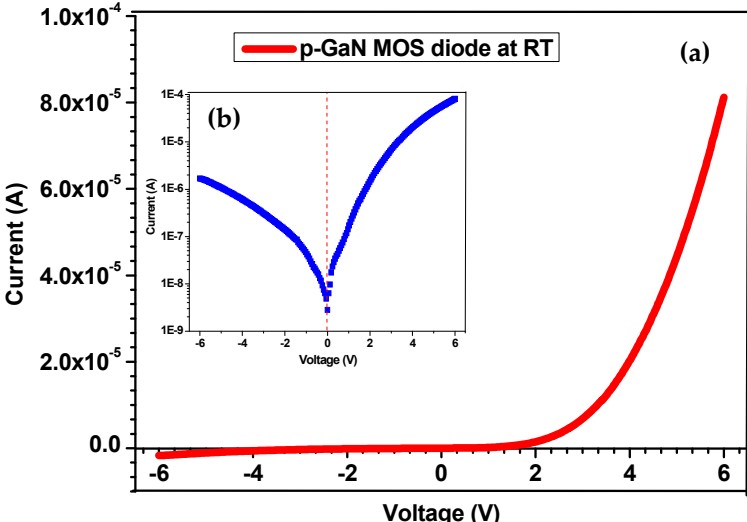

**Figure 3.** (**a**) *I–V* plot of the as-deposited Al/SiO$_2$/*p*-GaN MOS Schottky diode tested at RT, (**b**) the forward and reverse ln*I–V* characteristics of diodes.

According to the thermionic emission (TE) mode (for $qV > 3\,$kT), the electrical properties of the Schottky diode can be described as [6,11,22]:

$$I = I_0 \exp[q(V - IR_s)/n\mathrm{k}T] \tag{1}$$

The SBH can be expressed by [1,5,21]:

$$\phi_B = \frac{\mathrm{k}T}{q} \ln\!\left(\frac{AA^*T^2}{I_0}\right) \tag{2}$$

where $I_0$ is the saturation current, $V$ is the applied voltage, $R_s$ is the series resistance, $n$ is the ideality factor, $T$ is the measured temperature in Kelvin, $q$ is the electronic charge, k is the Boltzmann constant, $\phi_B$ is the Schottky barrier height (SBH), $A^*$ is the Richardson constant, $A$ is the contact area of the diode, and $A^*$ is the effective. The saturation current $I_0$ was defined by the intersection between the interpolated straight lines of the linear region and the current axis.

Using a stainless-steel mask with a square opening, the electrodes of our diode were measured at 1 mm$^2$. The $A^*$ value was 26.4 A·cm$^{-2}$·K$^{-2}$ (based on effective mass $m^* = 0.22 \times m_e$ for GaN, $m_e$ is electron mass) [4,5,13]. The ideality factor ($n$) from Equation (1) can be determined by [5,21,23]:

$$n = \frac{q}{\mathrm{k}T}\left(\frac{\mathrm{d}V}{\mathrm{d}(\ln I)}\right) \tag{3}$$

Based upon Equations (1) and (2), the SBH of the diode was 0.67 V, while the ideality factor $n$, based on Equation (3), was 3.32. According to Cheung's method, the series resistance $R_s$ and ideality factor can be found by the intersecting slope from the linear region of the d$V$/d(ln$I$) vs. the $I$ plots [11,22,24–26]:

$$\frac{\mathrm{d}V}{\mathrm{d}(\ln I)} = \frac{n\mathrm{k}T}{q} + IR_S \tag{4}$$

As shown in Figure 4, a calculation based on Equation (4) showed that the values of $R_s$ and $n$ were 5914 Ω and 3.51, respectively. Our MOS Schottky diode had high series resistance because there was an insulator SiO$_2$ layer of 70 nm between the metal and semiconductor.

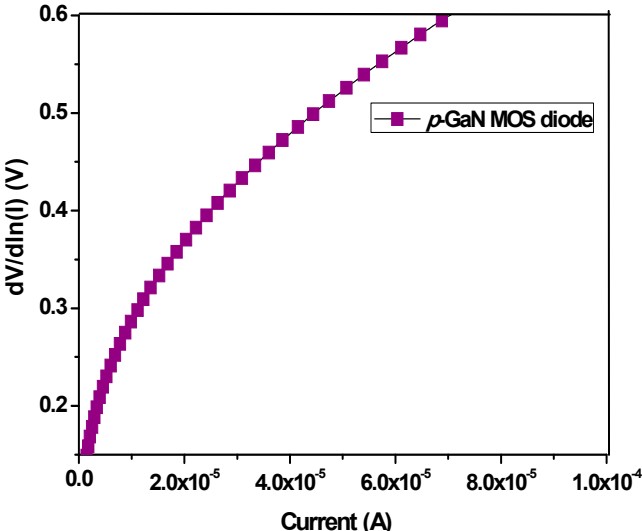

**Figure 4.** Plot of d*V*/dln(*I*) versus *I* for the as-deposited Al/SiO₂/*p*-GaN MOS Schottky diode.

The Norde method was also used to calculate the effective SBH of the diodes. The Norde function is described as the F(*V*, *I*) vs. the voltage *V*. It is given by [6,27]:

$$F(V, I) = \frac{V}{\gamma} - \frac{kT}{q} \ln\left(\frac{I}{AA^*T^2}\right) \tag{5}$$

The effective SBH $\phi_B$ is obtained by:

$$\phi_B = F(V_{min}) + \frac{V_{min}}{\gamma} - \frac{kT}{q} \tag{6}$$

where $\gamma$ is the first integer (dimensionless) is higher than *n*, F($V_{min}$) is the min value of F(*V*), and $V_{min}$ is the corresponding voltage [27,28].

Figure 5 displays the plot of F(*V*) vs. the *V* of the Al/SiO₂/*p*-GaN MOS Schottky diode measured at RT. Based on Equations (5) and (6), the SBH value was 0.78 eV for the device. Table 1 lists all the parameters of the diode, calculated by *I*–*V* test, and Cheung's and Norde's methods.

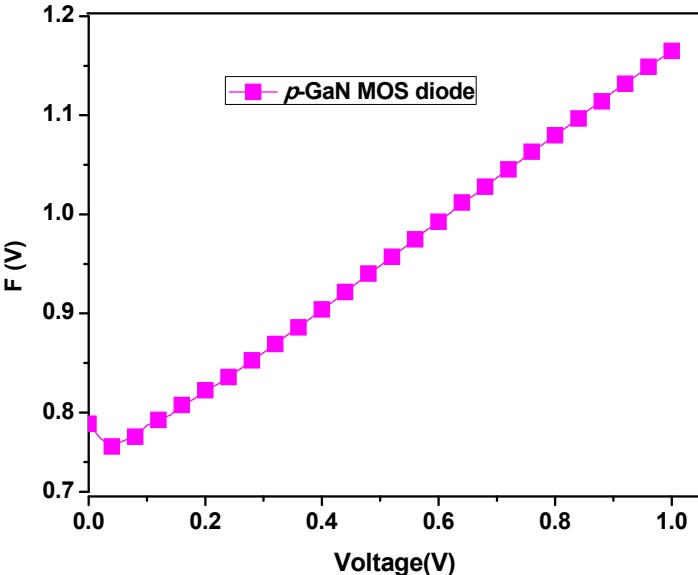

**Figure 5.** Characterization of the F(*V*, *I*) vs. *V* for the as-deposited Al/SiO₂/*p*-GaN MOS Schottky diode.

**Table 1.** The parameters of the *I–V* characteristics of the Al/SiO$_2$/*p*-MOS Schottky diode at room temperature.

| Sample | Leakage Current (A) at −1 (V) | Schottky Barrier Height (SBH) (eV) | | From *I–V* | Cheungs' Function d*V*/dln(*I*)–*I* | |
|---|---|---|---|---|---|---|
| | | *I–V* | Norde | *n* | *R$_s$* (Ω) | *n* |
| As-dep. | $4.49 \times 10^{-8}$ | 0.67 | 0.78 | 3.32 | 5914 | 3.51 |

### 3.3. Capacitance–Voltage (C–V) Characteristics

The capacitance–voltage (*C–V*) measurement of our diode was expressed and tested at room temperature, with a frequency range of 100 kHz–1 MHz. The *C–V* relationship of diodes can be expressed by [4,9,13]:

$$\frac{1}{C^2} = \frac{2(V_{bi} - \frac{kT}{q} - V)}{q\varepsilon_s N_p A^2}$$

(7)

$$N_p = \frac{2}{q\varepsilon_s A^2}\left[-\frac{1}{\mathrm{d}(1/C^{-2})/\mathrm{d}V}\right]$$

(8)

where $N_p$ is hole concentration, $V$ is the flat band voltage, $A$ is the area of the diode, and $\varepsilon_s$ is the permittivity of the semiconductor ($\varepsilon_s = 9.5 \times \varepsilon_o$ for GaN, $\varepsilon_o$ is electric constant) [4,13]. $V_0$ is determined by the plot of $1/C^2$ vs. $V$. The potential $V_{bi}$ is calculated from $V_0$ by [4,5,11]:

$$V_{bi} = V_0 + \frac{kT}{q}$$

(9)

The SBH $\phi_{CV}$ from the *C–V* measurement is given by [13,14,18]:

$$\phi_{CV} = V_{bi} + E_F - \Delta\Phi_b$$

(10)

where $E_F$ is the energy of Fermi level. This is given by [9,11,13]:

$$E_F = \frac{kT}{q}\ln\left(\frac{N_c}{N_p}\right)$$

(11)

Based on the $m^* = 0.22 \times m_e$ for GaN, $N_c$ is the density of states in the conduction band edge. It is expressed by [2,5,7]:

$$N_c = 2\left(\frac{2\pi m^* kT}{h^2}\right)^{3/2}$$

(12)

where $h$ is Plank constant. The $\Delta\Phi_b$ is the image force-induced barrier lowering. It is given by [7,9,13]:

$$\Delta\Phi_b = \left[\frac{qE_m}{4\pi\varepsilon_s\varepsilon_0}\right]^{1/2}$$

(13)

where $E_m$ is the maximum electric field and given by [9,13]:

$$E_m = \left[\frac{2qN_p V_0}{\varepsilon_s\varepsilon_0}\right]^{1/2}$$

(14)

Figure 6a shows the plotted *C–V* measurement of the MOS Schottky diode tested at the frequency range 100 KHz–1 MHz. The Figure 6b is the electrical properties of the diode, which was measured at a frequency of 1 MHz with an alternating current (AC) modulation of 100 mV. Figure 7 shows the characterization of $1/C^2$ vs. $V$ as a function of the *p*-MOS Schottky diode tested at different frequencies.

The *x*-intercept of the $1/C^2$ vs. *V* plot determined $V_0$ from the straight lines for the downward region at the reverse bias [4,5,7,11].

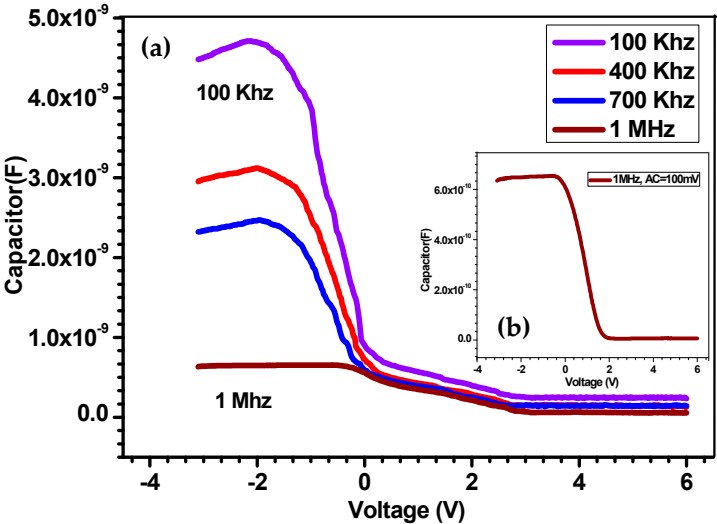

**Figure 6.** (**a**) Plot *C*–*V* measurement for the as-deposited Al/SiO$_2$/*p*-GaN MOS Schottky diode measurement at different frequencies between 100 kHz and 1 MHz, AC = 100 mV. (**b**) The electrical properties of the diode was measured at frequency of 1 MHz, AC = 100 mV.

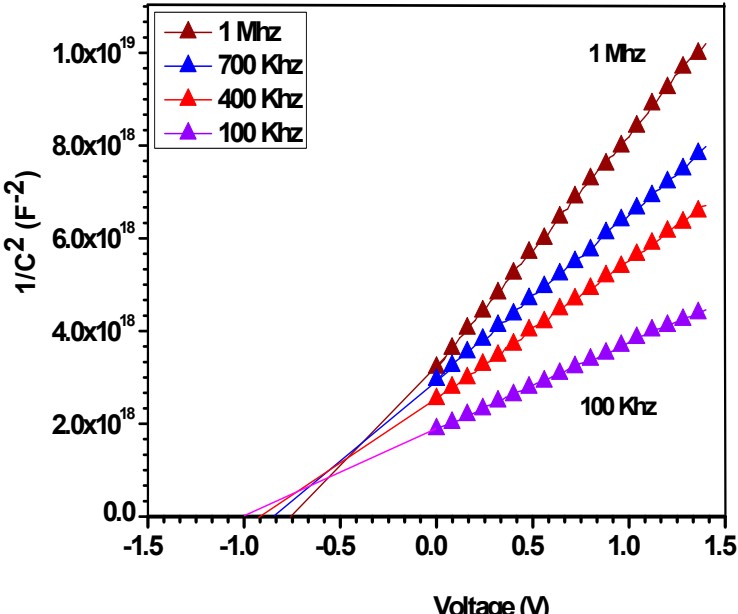

**Figure 7.** Plot of the $1/C^{-2}$–*V* for the as-deposited Al/SiO$_2$/*p*-GaN MOS Schottky diode measurement at different frequencies between 100 kHz and 1 MHz, AC = 100 mV.

Based on Equation (8), the hole concentration of the diode increased from $3.92 \times 10^{16}$ cm$^{-3}$ to $5.36 \times 10^{16}$ cm$^{-3}$ tested at 100 KHz to 1 MHz, while the Fermi level of the diode slightly decreased from 0.109 to 0.099 eV. After calculating the values of $V_0$, $V_{bi}$, $E_F$, $E_m$, and $\Delta\Phi_b$, based on Equations (9)–(14), the values of $V_0$ and the SBH values were reduced from 0.99 to 0.75 eV and 1.06 to 0.88 eV, respectively, when the testing frequencies were changed from 100 kHz to 1 MHz (Table 2).

**Table 2.** The parameters calculated from $1/C^2$–$V$ for characteristics of the Al/SiO$_2$/$p$-GaN MOS Schottky diode between 100 kHz and 1 MHz at the room temperature.

| Frequency (KHz) | $N_p$ (cm$^{-3}$) | $E_F$ (eV) | $V_0$ (eV) | $\Delta\Phi_b$ (eV) | $\phi_{CV}$ (eV) |
|---|---|---|---|---|---|
| 1000 | $3.92 \times 10^{16}$ | 0.109 | 0.75 | 0.039 | 0.88 |
| 700 | $4.20 \times 10^{16}$ | 0.106 | 0.82 | 0.042 | 0.94 |
| 400 | $4.89 \times 10^{16}$ | 0.103 | 0.90 | 0.044 | 1.02 |
| 100 | $5.36 \times 10^{16}$ | 0.099 | 0.99 | 0.046 | 1.06 |

## 4. Discussion

Tables 1 and 2 display all the parameters of the MOS Schottky diode measured by $I$–$V$ and $C$–$V$ measurements. This MOS Schottky diode showed an acceptable leakage current of ~$4.49 \times 10^{-8}$ A at −1 V, a breakdown voltage of −6 V, and a turn-on voltage of ~2.1 V. Calculating using Cheung's method, the MOS Schottky had $R_s$ of 5918 Ω and $n$ of 3.51. In addition, from the plot of the $I$–$V$ curve, the ideality factor $n$ was found to be 3.32. This indicated that a higher turn-on voltage leads to a higher ideality factor [4,9,13]. The growth of an insulator SiO$_2$ layer can effect to the accumulation layer during the forward bias, which affected to the high value of $R_s$ in our Schotky diode. For similar results, by measurement at RT, the series resistance ($R_s$) and ideality factor ($n$) were calculated to be 84.4 kΩ and 2.96 for the Pt/SiO$_2$/$n$-GaN MOS Schottky diode [6], 230 Ω and 1.6 for the Au/SiO$_2$/$n$-GaN MOS diode [13], and 66 Ω and 2.48 for the Al/SnO$_2$/$p$-Si (111) MOS diode, respectively [18].

From Table 2, we showed that the $C$–$V$ measurement data depended strongly on the tested frequency at the RT. At the high frequency (1 MHz), the interface state density could not identify the value of capacitance because it balanced with the semiconductor. At the low frequency, the interface state's density easily followed the AC signal. This created a signal and extra capacitance [11,13,18]. The $N_p$ and SBH decreased from $5.36 \times 10^{16}$ cm$^3$ to $3.92 \times 10^{16}$ cm$^3$ and from 1.06 to 0.88 eV, respectively, with increased frequency, due to the existence of the interfacial SiO$_2$ layer in the depletion region. Similarly, the $N_p$ of our Schottky diode was $5.36 \times 10^{16}$ cm$^{-3}$ when tested at a frequency of 1 MHz at RT [7]. The $N_p$ and SBH of the Au/SiO$_2$/$n$-GaN MOS diode were $2.08 \times 10^{17}$ cm$^{-3}$ and 0.99 eV, also tested at a frequency of 1 MHz [9].

The hole's concentration ($N_p$), calculated from the $1/C^2$–$V$ plots for the MOS Schottky diode, was lower than that calculated by the Hall measurement because this result was measured from the $p$-Mg-GaN film deposited on the SiO$_2$/Si substrate. In addition, with the fast growth rate in deposition, the interface between $p$-GaN and SiO$_2$ affected the polarized SiO$_2$ layer. The electrical field across the depletion region changed significantly near the $p$-GaN layer. It was affected by the strong variation in the hole's concentration, leakage current, and turn-on and breakdown voltages of the diode [11,15,16].

Therefore, the SBH of our MOS Schottky diode, in terms of $C$–$V$ measurement, was higher than the $I$–$V$ test because of the charges induced by dielectric layer. The SBH of our diode corresponded with some results (0.67–1.06 eV) of the GaN Schottky diodes made by metal organic chemical vapor deposition (MOCVD) and other approaches. Cheng et al. also reported the SBH of a Pt-oxide-Al$_{0.3}$Ga$_{0.7}$As MOS diode, which decreased from 1.03 to 0.86 eV after annealing in a hydrogen atmosphere [16]. Baris et al. reported all the parameters of Au/TiO/$n$-Si (100) MOS diodes; an ideality factor of 3.72 and an SBH of 0.62 eV were determined by testing the $I$–$V$ measurement. Meanwhile, via calculation by $C$–$V$ measurement, the SBH and bulk concentration were determined to be 0.99 eV and $9.82 \times 10^{14}$ cm$^{-3}$, respectively [23].

## 5. Conclusions

The modeling and electrical properties of the Al/SiO$_2$/$p$-GaN MOS Schottky diode were successfully established by total RF sputtering. All the parameters were calculated based on $I$–$V$ and $C$–$V$ measurements. The SBHs of the MOS Schottky diodes were determined to be 0.67 ($I$–$V$), 0.78 eV (Norde), and 0.88 eV ($C$–$V$). The hole's concentration, tested by $C$–$V$ measurement, decreased slightly

compared with that determined by the Hall measurement. This was due to the existence of the $SiO_2$ layer in the *p*-GaN MOS diode. Our work using cost-effective RF sputtering to make the Al/$SiO_2$/*p*-GaN MOS Schottky diode can be applied to the development of electronic devices.

**Author Contributions:** Data curation, P.T.C. and T.A.T.T.; methodology, writing—original draft, investigation, P.T.C. and T.A.T.T., formal analysis, funding acquisition, writing—review and editing, P.T.C., T.A.T.T., P.Q.-P., V.K.N. and N.P.L.T.; supervision, D.-H.K.

**Funding:** This research was funded by the Ministry of Science and Technology of the Republic of China under grant number 107-2221-E-011-141-MY3.

**Conflicts of Interest:** The authors declare no conflict of interest.

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
