# Peer review of "Electrical and Structural Properties of All-Sputtered Al/SiO2/p-GaN MOS Schottky Diode"

_coatings, doi:10.3390/coatings9100685_

Round 1
Reviewer 1 Report
This paper investigated the nature of Al/SiO2/p-GaN MOS Schottky diode formed by RF-sputtering. Carrier transport mechanism was investigated based on TE model. The results suggest various diode parameters without much meaning. The reviewer suggests denying this manuscript for publication in Coatings
==================================================================
It is better to add a background sentence at the beginning of the Abstract. At the same time, the abstract part needs to be more concise. Too lengthy.
The English in this work need polishing. The expression is not academic enough and sometimes grammatical and even formatted issues can be found. Please carefully check and revise them.
The paper has to give XRD information of p-GaN. The structure in this work used p-GaN on insulator, which result in the formation of polycrystalline phase or amorphous phase of GaN, not single crystalline. Using this phase, the reviewer believe that there is no reason to use this structure in the device since the performance will be significantly suppressed.
The article insisted that he MOS Schottky diode had remarkable improvements in the leakage current. Compare to what??
The current transport mechanism in the diodes has to be deciphered in various temperature range. Please use T-I-V measurement. Ideality factor, schottky barrier height at only RT?? What do they mean on earth? They are just useless lists. Also, TE model is not proper to explain the tunneling characteristics. Other mechanism, such as TFE model has to be included.
Author Response
Dear reviewer, thanks you for your comment
Answers:
Dear reviewer, thanks you for your comment,
We will re-write abstract and edit English all manuscripts.
However, some your evaluations are very negative and unfair for reviewing in research science. Fox example:
- what is improve?
- Ideality factor, schottky barrier height at only RT ?? What do they mean on earth? They are just useless lists….
So, we must refuse to answer some the requirements, and we also share some information:
- Regarding about XRN of p-GaN, we reported in previous work, that had published in some SCI papers (please refer on list below). In this paper, we want to the design the p-GaN MOS Schottky diode, and from that, we test and compare all parameters of p-MOS Schottky diode by I-V and C-V measurement at room temperature and different frequencies
- You said that “what is improve”?, we cheked abstract, and the paper showed that the improvement of leakage current by comparison with some previous publications about on this.
- Ideality factor, schottky barrier height at only RT ?? What do they mean on earth? They are just useless lists.
Dear reviewer,
These seems that you are not interested in this field, we investigated Schottky diode and tested electrical properties at room temperature, and compared the Schottky barrier height and other parameter by using Norde and Cheungs’ method. TE mode was used very widely and we had a lot of Success in applying this model.
You can refer some our publication about Schottky diode and p-n junction diode. Thanks you!
1/ T.T.A.Tuan, D.H.Kuo, C.C. Li, W.C.Yen Characteristics of RF reactive sputter-deposited Pt/SiO2/n-InGaN MOS Schottky diodes, J. Mater. Sci. Semicond. Process. 30 (2015) 314–320. https://doi.org/10.1016/j.mssp.2014.10.021
2/ T.T.A.Tuan, D.H.Kuo, Temperature-dependent electrical properties of the sputtering-made n-InGaN/p-GaN junction diode with a breakdown voltage above 20 V .https://doi.org/10.1016/j.mssp.2015.01.011
3/ C.C. Li, D.H. Kuo, Material and technology developments of the totally sputtering-made p/n GaN diodes for cost-effective power electronics, J. Mater. Sci. Mater. Electron. 25 (2014) 1942-1948. DOI: 10.1007/s10854-014-1826-1
4/ D.H.Kuo, T.T.A.Tuan, C.C. Li, W.C.Yen, Electrical and structural properties of Mg-doped InxGa1−xN (x ≤ 0.1)and p-InGaN/n-GaN junction diode made all by RF reactive sputtering, Mater.Sci.Eng., B 193 (2015) 13–19. https://doi.org/10.1016/j.mseb.2014.11.005
5/ T.T.A.Tuan, D.H.Kuo, Temperature-dependent electrical properties of the sputtering-made n-InGaN/p-GaN junction diode with a breakdown voltage above 20 V .https://doi.org/10.1016/j.mssp.2015.01.011
6/ C.C. Li, D.H. Kuo, Material and technology developments of the totally sputtering-made p/n GaN diodes for cost-effective power electronics, J. Mater. Sci. Mater. Electron. 25 (2014) 1942-1948. DOI: 10.1007/s10854-014-1826-1
Reviewer 2 Report
This work is about “Electrical and structural properties of all-sputtered Al/SiO2/p-GaN MOS Schottky diode”. The results of this paper are interesting, which would attract attentions from a wide range of readers. However, there are still some ambiguous. Thus I recommend it to be accepted after some minor revisions. Some suggestion and comments are listed below
More precise and to the point abstract and conclusions Introduction can be elaborated with significance and more literature review for different aspects. Authors are requested to produce a comparative chart to correlate current results with the published work. The authors should check the whole paper carefully for many mistakes in grammar and format.
Author Response
Dear reviewer, thanks you for your comment
Reviewer 3 Report
Authors report on "all sputtering-made Al/SiO2/p–GaN MOS Schottky diode" fabricated using radio–frequency (RF) reactive sputtering technique. Th paper contains experimental data proving advantages of used techniques and fabricated device. Theoretical model supporting the experimental data and there interpretation is provided. The paper will be of interest for a broad community working on design and fabrication of novel semiconductor devices.
The tex is full of typos and the English language should proof read before publishing. Just one example: "The electrical properties of MOS Schottky diode was studied by I–V and C–V."
Author Response
Dear reviewer, thanks you for your comment
Dear reviewer,
Thanks you for your comments
We will re-write abstract, conclusions, introduction and edit English all manuscripts.
Round 2
Author Response
Dear reviewer, thanks you for your comment. we answer by a point-by-point response

Round 3
Reviewer 1 Report
The reviewer agree to accept this article in the current form.